# IL-33 drives group 2 innate lymphoid cell-mediated protection during *Clostridium difficile* infection

Alyse L. Frisbee [1], Mahmoud M. Saleh[1], Mary K. Young [2], Jhansi L. Leslie[2], Morgan E. Simpson[3], Mayuresh M. Abhyankar[2], Carrie A. Cowardin[1], Jennie Z. Ma[4], Patcharin Pramoonjago[3], Stephen D. Turner[4], Alice P. Liou[5], Erica L. Buonomo[1] & William A. Petri Jr.[1,2,3]

*Clostridium difficile* (*C. difficile*) incidence has tripled over the past 15 years and is attributed to the emergence of hypervirulent strains. While it is clear that *C. difficile* toxins cause damaging colonic inflammation, the immune mechanisms protecting from tissue damage require further investigation. Through a transcriptome analysis, we identify IL-33 as an immune target upregulated in response to hypervirulent *C. difficile*. We demonstrate that IL-33 prevents *C. difficile*-associated mortality and epithelial disruption independently of bacterial burden or toxin expression. IL-33 drives colonic group 2 innate lymphoid cell (ILC2) activation during infection and IL-33 activated ILC2s are sufficient to prevent disease. Furthermore, intestinal IL-33 expression is regulated by the microbiota as fecal microbiota transplantation (FMT) rescues antibiotic-associated depletion of IL-33. Lastly, dysregulated IL-33 signaling via the decoy receptor, sST2, predicts *C. difficile*-associated mortality in human patients. Thus, IL-33 signaling to ILC2s is an important mechanism of defense from *C. difficile* colitis.

[1] Department of Microbiology, Immunology and Cancer Biology, University of Virginia Health System, Charlottesville, VA 22908, USA. [2] Department of Medicine, University of Virginia Health System, Charlottesville, VA 22908, USA. [3] Department of Pathology, University of Virginia Health System, Charlottesville, VA 22908, USA. [4] Department of Public Health Sciences, University of Virginia School of Medicine, Charlottesville, VA 22908, USA. [5] Seres Therapeutics, Cambridge 02139 MA, USA. Correspondence and requests for materials should be addressed to W.A.P.Jr. (email: wap3g@virginia.edu)

Clostridium difficile (C. difficile) is an anaerobic, spore-forming bacterium, that is a growing public health threat. The emergence of hypervirulent NAP1/027 isolates over the past decade have caused worldwide epidemics with high mortality rates[1]. In the United States alone, an estimated half a million C. difficile cases and 29,000 deaths occurred in 2011[2]. Disruption of the intestinal microbiota, most often through antibiotic exposure, greatly increases the risk of C. difficile infection of the colon. The antibiotics vancomycin, fidaxomicin, and metronidazole are the current standard of care to treat C. difficile infection (CDI); however, recurrence occurs in up to 35% of patients[3]. Microbial strategies to combat severe C. difficile colitis through fecal microbiota transplantation (FMT) have seen great advancements in recent years[4]. However, how FMT strategies alter mucosal immune defenses and how the host immune system contributes to C. difficile disease pathogenesis are not fully understood.

Hypervirulent NAP1/027 clones, such as the epidemic R20291 strain used in this study, are associated with higher mortality rates, express higher amounts of the classical C. difficile exotoxins, Toxin A and Toxin B, and also produce a third toxin called C. difficile transferase (CDT)[5]. All three toxins can together cause symptomatic disease by damaging colonic epithelial cells and eliciting pro-inflammatory, inflammasome-dependent IL-1β release[6,7]. Development of a murine model of C. difficile colitis has allowed for better understanding of disease onset, which is characterized by an acute 2–5 day inflammatory response with high neutrophil influx, diarrhea, and weight loss followed by a resolution phase with weight-gain[8,9]. Importantly, it has also led to new insights into the relationship between the host immune response and disease development. Recent studies in both mice and humans have demonstrated an important yet multifaceted role for the host immune system during C. difficile infection. Early acute cellular responses, such as MYD88-mediated neutrophilia, IL-25-responsive eosinophils, and IFN-γ+ producing group 1 innate lymphoid cells (ILC1s), provide protection during C. difficile infection[10–12]. However, many pro-inflammatory pathways such as Th-17-associated IL-23, IL-17, and toll-like receptor 2 (TLR2) signaling exacerbate C. difficile-associated mortality[6,13–15]. Several of these murine studies have been corroborated in human patients with IFN-γ, IL-5 and peripheral eosinophil counts being associated with less severe infection yet pro-inflammatory cytokines, such as IL-8, IL-2, and IL-6 being associated with poor prognosis[15–18]. These studies emphasize the complexity of the immune network elicited during CDI. Likely, a robust antimicrobial response is vital for bacterial clearance, however, this circuit must be tightly regulated to prevent chronic intestinal damage. Thus, further investigation is required to discover immune avenues that control CDI-associated tissue damage.

In this study, we use whole-tissue transcriptomics to identify potential therapeutic immune pathways that aid in recovery during infection with the hypervirulent strain, R20291. We identify the IL-1 cytokine family member, IL-33, as an important guardian of the gut barrier during C. difficile colitis that prevents CDI-associated mortality via activation of group 2 innate lymphoid cells (ILC2s). We demonstrate a role for IL-33 in human CDI and show that dysregulation of IL-33 signaling is associated with higher patient mortality. Finally, we demonstrate a connection between fecal microbiota transplantation (FMT) therapy and IL-33 levels within the colon, indicating a potentially targetable approach for increasing colonic IL-33 with rationally designed, next-generation microbial cocktails. In summary, these studies demonstrate that IL-33 signaling to ILC2s is an essential pathway for recovery from C. difficile-associated colitis.

## Results

**Transcriptomics identifies IL-33 upregulation during CDI**. To study host-derived gene expression changes during hypervirulent CDI, we infected mice with the CDT expressing epidemic PCR-ribotype 027 strain, R20291, or with an attenuated isogenic mutant lacking the binding region of CDT (R20291_cdtb)[19]. As we previously reported, the CDTb mutant strain caused a less virulent infection with a reduction in mortality, weight loss, and clinical severity (Supplementary Fig. 1a–c), confirming CDT's role as a virulence factor[6]. To find possible therapeutic immune targets relevant during severe infection, we compared whole-cecal tissue transcriptomes of R20291 vs. R20291_cdtb-infected mice via Affymetrix microarray. In total, 484 genes were differentially expressed in response to severe CDT+ infection [logfc ≥ 0.5, P < 0.05] (Supplementary Data 1). The microarray analysis revealed a 1.8-fold increase in the alarmin and type-2 cytokine, interleukin-33 (IL-33) (Fig. 1a, b).

To assess the relevance of IL-33 during C. difficile infection, we quantified IL-33 protein and RNA during R20291 and R20291_cdtb infections and in uninfected controls. Both qPCR and ELISA analyses indicated upregulation of IL-33 in response to both hypervirulent R20291 and attenuated R20291_cdtb infection (Fig. 1c; Supplementary Fig. 1d). Mice infected with the hypervirulent strain had a 1.71-fold increase in IL-33 protein relative to R20291_cdtb-infected mice, and R20291_cdtb-infected mice had a 1.82-fold increase in IL-33 protein relative to uninfected controls. This data indicate that IL-33 expression is increased in response to increasing severity of C. difficile infection.

Further pathway analysis of the microarray data set using Ingenuity[20], and Consensus Path DB[21] pathway tools revealed enrichment of many immune-related pathways in response to the CDT expressing strain, including many tissue-regulatory pathways, including IL-10R, IL-4, and IL-13 (Supplementary Fig. 1e; Fig. 1d). These pathways have been associated with tissue repair and can counter pro-inflammatory responses to promote healing[22,23]. Given increased IL-33 expression in response to CDI, in addition to its established role in promoting IL-10, IL-4, and IL-13 tissue-regulatory pathways, we hypothesized IL-33 may be an important upstream regulator of gut-barrier defenses during C. difficile infection.

**IL-33 protects from C. difficile-associated disease**. In order to determine if IL-33 mediates protection during infection, we used a mouse model of CDI to ascertain the effect of IL-33 on disease severity, tissue pathology, and mortality[6,8,10,13]. First, we asked whether increasing IL-33 levels in the gut alters disease severity during infection. IL-33 was administered in five doses by intraperitoneal injection (0.75 µg/mouse) prior to infection (Supplementary Fig. 2a). Our IL-33 treatment regimen caused a fivefold increase in IL-33 protein levels within the colon prior to infection (Supplementary Fig. 2b). Notably, this IL-33 treatment regimen reduced mortality following infection with R20291 as evidenced by the increased survival (70% with IL-33 treatment vs. 30% without) (Fig. 2a). In addition, IL-33-treated mice had reduced weight loss, and less severe clinical scores (a measurement system of weight loss, piloerection, ocular discharge, activity, posture, and diarrhea) (Fig. 2b, c). We also noticed that IL-33 treatment reduced epithelial barrier disruption during infection. Blinded histology scoring of the epithelial barrier by H&E staining revealed a significant reduction in epithelial damage and submucosal edema (Fig. 2d, e). Accordingly, IL-33-treated mice had reduced gut permeability as quantified by FITC-dextran gut-barrier permeability assay (Supplementary Fig. 2c).

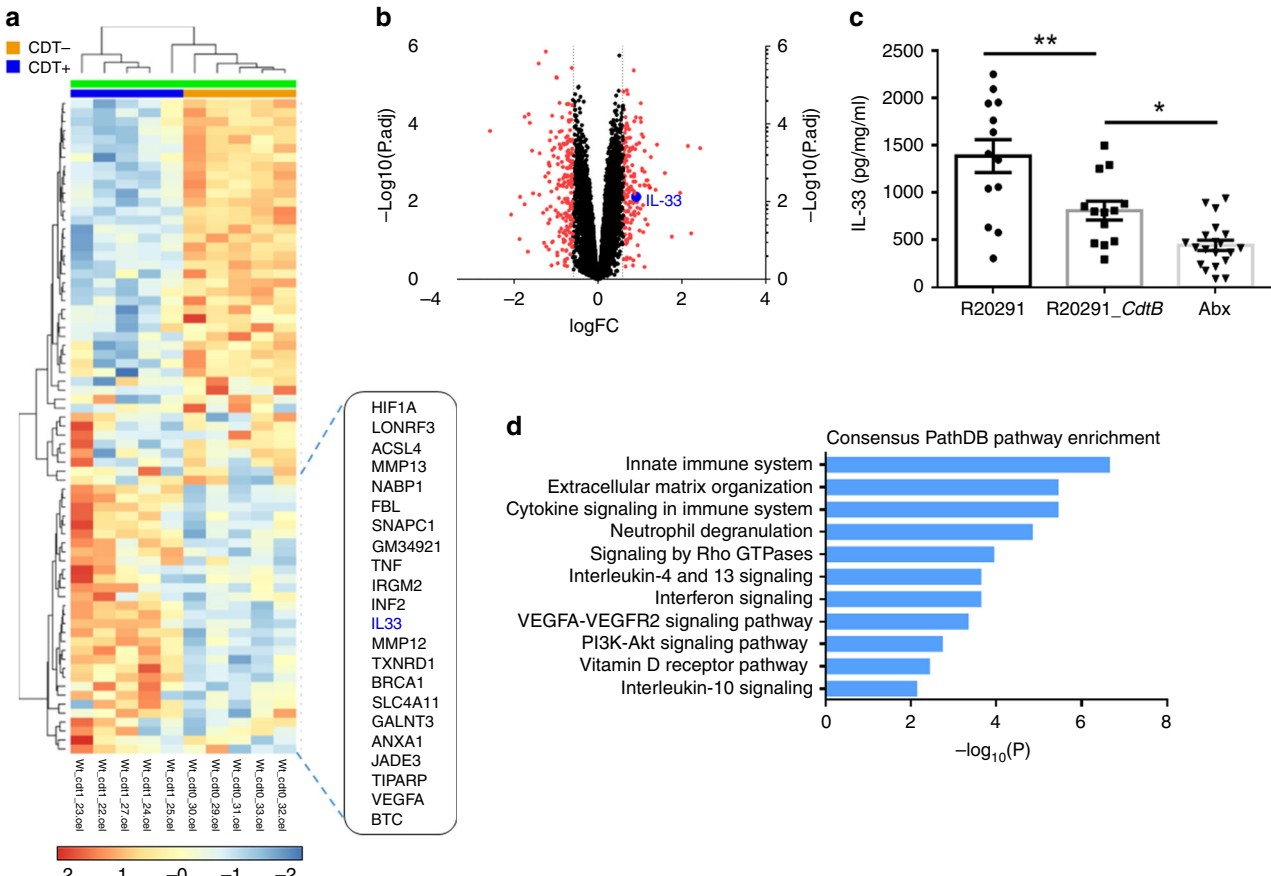

**Fig. 1** IL-33 is upregulated by host in response to *C. difficile* infection. Mice were infected with R20291 (CDT+) or attenuated R20291_cdtb (CDT-) and whole-cecal tissue transcriptomic analyses was performed on day 3 post infection. **a** Heatmap of genes upregulated (red) or downregulated (blue) in response to CDT toxin. **b** Volcano plot highlighting IL-33 (blue) among genes altered (>0.5 logFC = red). **c** ELISA of IL-33 protein within the cecal tissue (day 3) of R20291 infected, R20291_cdtb infected, or uninfected antibiotic-treated (ABX) mice. **d** Enriched pathways of the top upregulated transcripts (log FC > 0.5; $p < 0.05$) listed by significance ($-\log10(p)$) and created using the ConsensusPathDP database. (**a**, **b**, $n = 5$ biologically independent animals). **c** Comparison made by ANOVA ($n = 13, 13, 20$ combined from two independent experiments). Individual $n$ numbers in each panel are shown in same order as their corresponding groups from left to right, and this arrangement is applied to other relevant figures in the text. Statistical significance is demarked as *$P < 0.05$, **$P < 0.01$, and ***$P < 0.001$, throughout the text. Error bar indicates SEM

Despite the reduction in epithelial damage, IL-33-mediated protection was not due to a reduction in *C. difficile* bacterial burden as IL-33-treated mice were colonized to equivalent levels as vehicle controls (Fig. 2f). IL-33-mediated protection was also not caused by a reduction in toxins A/B or CDT, as IL-33 treatment mice had equivalent toxin levels over the course of acute infection when peak disease occurred (Supplementary Fig. 3a, b). Furthermore, IL-33 treatment did not prevent translocation of gut commensals as IL-33-treated mice had equivalent dissemination of commensal bacteria into their livers (Supplementary Fig. 2d). This is likely caused by gut leakiness still present in IL-33-treated mice, as demonstrated by FITC-dextran still present within their serum during infection (Supplementary Fig. 2c).

IL-33-mediated protection was not restricted to hypervirulent R20291 infection as IL-33 treatment robustly protected against the non-ribotype 027 *C. difficile* strain VPI 10463 (Supplementary Fig. 4a–c), and the attenuated R20291_cdtb mutant strain (Supplementary Fig. 4d–f).

Since the microbiota plays a critical role in the pathogenesis of CDI[24,25], we assessed whether IL-33 treatment altered the microbiota prior to infection via analysis of the V4 region 16S rRNA gene amplified from cecal contents of IL-33 vs. vehicle-treated mice. We found no significant differences in the alpha diversity or beta diversity between the treatment groups

(Supplementary Fig. 5a–c). Thus, IL-33 treatment did not significantly alter the microbiota composition prior to infection with *C. difficile*.

Given the striking protection elicited by exogenous IL-33 treatment, we additionally asked whether endogenous IL-33 signaling contributes to survival from CDI using transgenic mice lacking the IL-33 receptor, ST2[-/-]. Complementing the IL-33 treatment data, ST2[-/-] mice had higher mortality rates compared with cohoused wild-type controls (Fig. 2g). Markedly, ST2[-/-] mice lost more weight and developed clinical signs by day 1 of infection when wild-type mice lacked signs of disease (Fig. 2h, i). Similar to the IL-33 treatment data, ST2[-/-] mice did not have more severe disease due to increased *C. difficile* bacterial burden or increased toxin levels, as there were no significant differences in *C. difficile* colonization or toxins A/B, and CDT levels between ST2[-/-] vs. controls on day 2 post infection when disease was most severe (Fig. 2f; Supplementary Fig. 3c, d). Similar to the IL-33 treatment model, endogenous IL-33 protection was not restricted to binary toxin expressing strains as only 40% of ST2[-/-] mice survived attenuated R20291_cdtb infection, compared with 100% of wild-type mice (Supplementary Fig. 4g–i). Together these results indicate that IL-33 signaling protects from mortality during CDI independent of *C. difficile* bacterial burden and toxin expression.

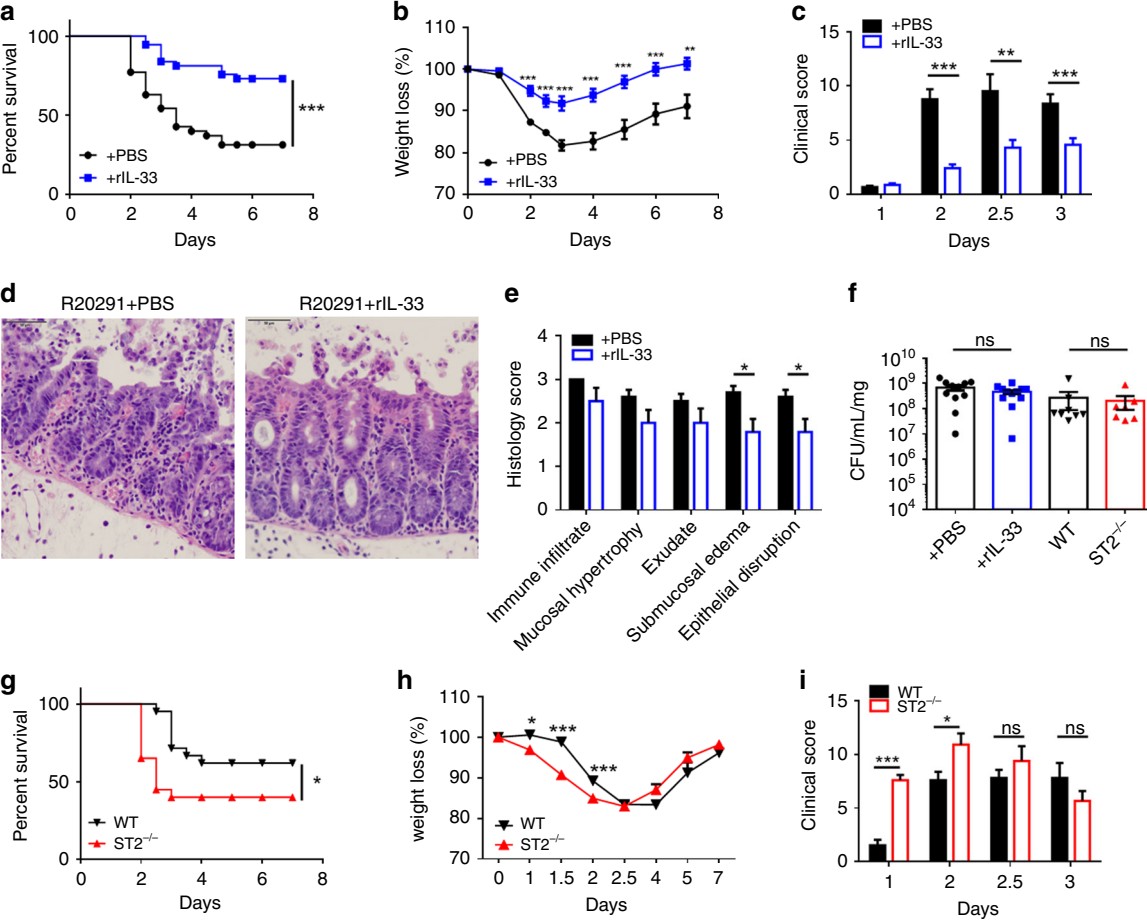

**Fig. 2** IL-33 protects from mortality and epithelial barrier disruption during *C. difficile* infection. C57BL6 mice (**a–f**) or ST2$^{-/-}$ mice (**f–i**) were treated with an antibiotic cocktail prior to infection with R20291 and treated by IP injection with recombinant IL-33 (rIL-33) or vehicle control (PBS) for 5 days prior to infection. **a** Survival curves (**b**), weight loss (**c**), and clinical scores after infection and treatment. (**d**, **e**) Infected cecal tissue was examined with IL-33 treatment and without (day 3). **d** Representative epithelial damage (H&E) of treatment groups assessed by (**e**) blinded scoring of infected tissue (H&E). **f** Peak *C. difficile* bacterial burden in cecal contents after infection (day 2) of IL-33-treated mice or ST2$^{-/-}$ mice compared with the vehicle or wild-type controls. **g–i** ST2$^{-/-}$ on C57BL6 background and C57BL6 controls were cohoused for 3 weeks prior to R20291 infection. **g** Survival curves (**h**) weight loss and (**i**) clinical scores were assessed. **a**, **g** Comparison made by log rank test (**a**, $n = 35, 37$; **g**, $n = 21, 20$). **b–e**, **h–i** Comparison made by Student's *t* test (**b**, $n = 39$; **c**, $n = 29, 30$; **e**, $n = 10$; **h**, $n = 27, 26$; **i**, $n = 20$). **f** Comparisons for *C. difficile* burden were made by Mann–Whitney U tests (**f**, $n = 11, 12, 8, 7$). **a–c**, **g–i** The data combined from three independent experiments. **d–f** The data representative of three independent experiments. Statistical significance is demarked as *$P < 0.05$, **$P < 0.01$, and ***$P < 0.001$. Scale bar is 50 μm. Error bar indicates SEM

**IL-33 regulates type-2 immunity during CDI**. In addition to the important role of *C. difficile* toxins in driving epithelial disruption and disease, the type of immune response generated during CDI is an important predictor of disease severity[18]. Colonic neutrophil invasion and Type-17 inflammation are hallmarks of CDI, whereas type-2-associated eosinophilia is protective in both human and murine CDI[13,15,16]. Given these findings, we wanted to determine whether IL-33 modulates the local intestinal immune response during infection to prevent mortality and epithelial disruption. Indeed, flow cytometry revealed that IL-33 treatment caused a dramatic switch in the myeloid cells of the colon by day 2 post infection, decreasing pro-inflammatory neutrophil and monocyte frequencies and numbers (Fig. 3b, c; Supplementary Fig. 6a, c, d). Instead of the canonical neutrophilic response seen in wild-type infection, IL-33 treatment shifted the myeloid compartment toward dominant eosinophilia by day 2 post infection (57% with IL-33 vs. 20% without) and a trend toward increased numbers of eosinophils (Fig. 3b, c; Supplementary Fig. 6a, b). This shift in myeloid cells occurred only after infection, as there were no differences in the frequency or

number of neutrophils, monocytes, or eosinophils in uninfected controls (Fig. 3a; Supplementary Fig. 6b–d).

In alignment with the IL-33 treatment model, ST2$^{-/-}$ mice, which lack IL-33 signaling, had a reduction in colonic eosinophils both by frequency and number compared with wild-type controls, indicating that endogenous IL-33 signaling contributes to eosinophil accumulation within the colon during infection (Supplementary Fig. 6f, g). Unlike the IL-33 treatment model, ST2$^{-/-}$ knockout mice had no change in colon neutrophilia during CDI, indicating endogenous IL-33 dominantly regulates colonic eosinophil recruitment during infection (Supplementary Fig. 6h, i).

Given the IL-33-mediated switch toward the gut eosinophilia, we questioned whether IL-33 treatment also altered the cytokine milieu during CDI. We found that IL-33 treatment reduced pro-inflammatory IL-1β, IL-6, and IL-23 and increased anti-inflammatory IL-4, IL-5, and IL-10 during CDI (Fig. 3d, e). In accordance with the myeloid cell data, the increase in the type-2-associated cytokine, IL-4, occurred only after infection as uninfected IL-33-treated mice had equivalent levels of IL-4

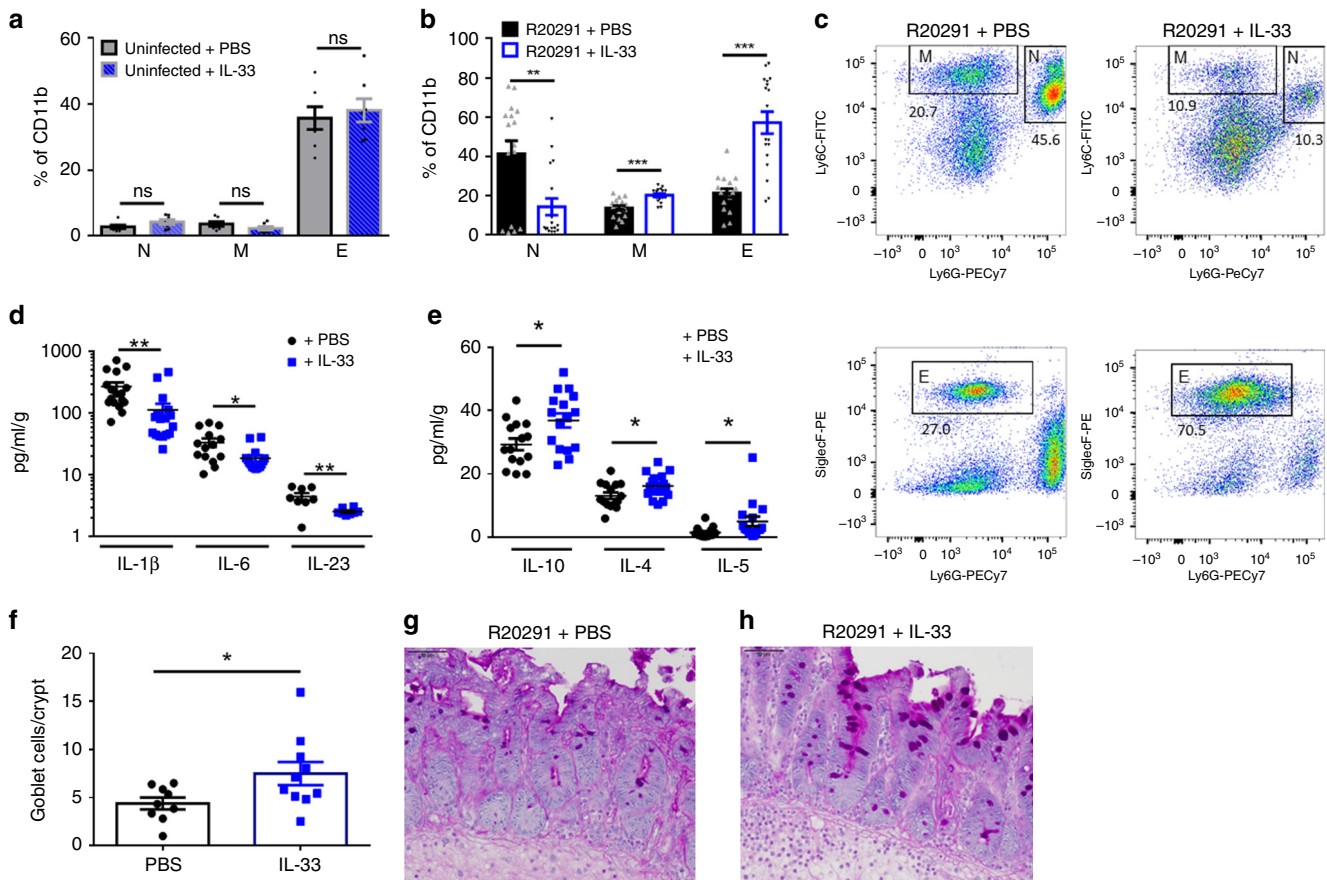

**Fig. 3** IL-33 dampens inflammation and skews gut type-2 immunity during CDI. **a–c** Immune profiling of myeloid cells within the colon of IL-33-treated or untreated mice. **a**, **b** Quantification of the frequency of N = neutrophils (CD45+ CD11b+ Ly6g+ Ly6c+), M = monocytes (CD45+ CD11b+ Ly6g- Ly6c+), and E = eosinophils (CD45+ CD11b+ SiglecF+ Ly6g−) in the colon of (**a**) antibiotic-treated mice or (**b**) R20291-infected mice. **c** Representative dot plots of myeloid cells within the colon of IL-33-treated (right plots) vs. untreated (left plots) mice. **d**, **e** Cytokine protein expression in cecal tissue during *C. difficile* infection. **d** Proinflammatory IL-1β, IL-6, and IL-23 expression and (**e**) anti-inflammatory IL-10, IL-4, and IL-5 expression with and without IL-33 treatment. **f–h** Enumeration of mucin+ goblet cells with and without IL-33 treatment by microscopy. **f** Blinded quantification of number of PAS+ goblet cells per colon crypt during R20291 infection (day 2). **g**, **h** Representative PAS+ goblet cell staining of the (**g**) vehicle or (**h**) IL-33 treated mice. **a–e** Comparison made by two-tailed student t test as described in methods (**a**, *n* = 7; **b**, *n* = 17,18, **d**, *n* = 17,16,8; **e**, *n* = 15,16; **f**, *n* = 9,10). The data representative of three independent experiments. **b–e** Two combined independent experiments. Statistical significance is demarked as *P < 0.05, **P < 0.01, and ***P < 0.001. Scale bar is 50 μm. Error bar indicates SEM

relative to controls (Supplementary Fig. 6e). In addition to eosinophilia, type-2-associated mucosal immunity is linked to goblet cell expansion to aid in parasite expulsion and is regulated by IL-13 production from type-2 innate lymphoid cells in the gut[26,27]. CDI causes a loss of goblet cells and mucin production at the epithelial barrier of the colon[28,29]. Thus, we explored the effects of IL-33 treatment on goblet cell mucin responses by periodic acid Schiff (PAS+) goblet cell mucin staining of infected cecal sections. IL-33-treated mice had significantly more goblet cells per colonic crypt during CDI infection compared with untreated controls (Fig. 3f–h). Taken together, these results indicate that IL-33 is an important regulator of the balance between type-17 vs. type-2-associated mucosal immunity during CDI.

**IL-33 activated ILC2s protect from CDI-associated disease**. IL-33 has been shown to reduce intestinal damage during IBD colitis via the action of type-2 innate lymphoid cells (ILC2s) or ST2+ regulatory T cells[30,31]. Furthermore, B cells have also been implicated in IL-33-mediated protection from IBD colitis via IgA or IL-10 production[32,33]. We wondered whether innate lymphoid cells (ILCs) or adaptive T and B cells were involved in the

therapeutic capacity of IL-33 to protect from hypervirulent *C. difficile* colitis. To address this question, we treated Rag2−/− (lacking T and B cells) and Rag2−/− γc−/− (additionally lacking ILCs) mice with IL-33 or the vehicle control. In line with Abt et al., Rag2−/− γc−/− mice had significantly higher mortality compared with Rag2−/− mice (75% Rag2−/−γc−/− vs. 24% Rag2−/−) confirming the importance of ILCs in recovery from *C. difficile*[11]. IL-33 treatment of Rag2−/− mice led to a significant increase in survival and reduction in weight loss, indicating T and B cells are dispensable for IL-33-mediated protection during CDI (Fig. 4a, b).

In contrast, IL-33-mediated protection from mortality and weight loss was lost in Rag2−/−γc−/− mice, indicating that ILCs are necessary for IL-33's therapeutic effects (Fig. 4a, c). We found that during CDI, colonic GATA3+ ILC2s co-expressed the IL-33 receptor, ST2, leading us to believe that ILC2s were downstream of IL-33-mediated protection (Supplementary Fig. 7a, b). Indeed, protective IL-33 treatment led to a significant increase in ST2+ ILC2s within the colon lamina propria during infection (Fig. 4d, e; Supplementary Fig. 7c). Furthermore, IL-33-treated Rag2−/− mice had an equivalent increase in the number of colonic ILC2s, indicating that IL-33-mediated ILC2 induction

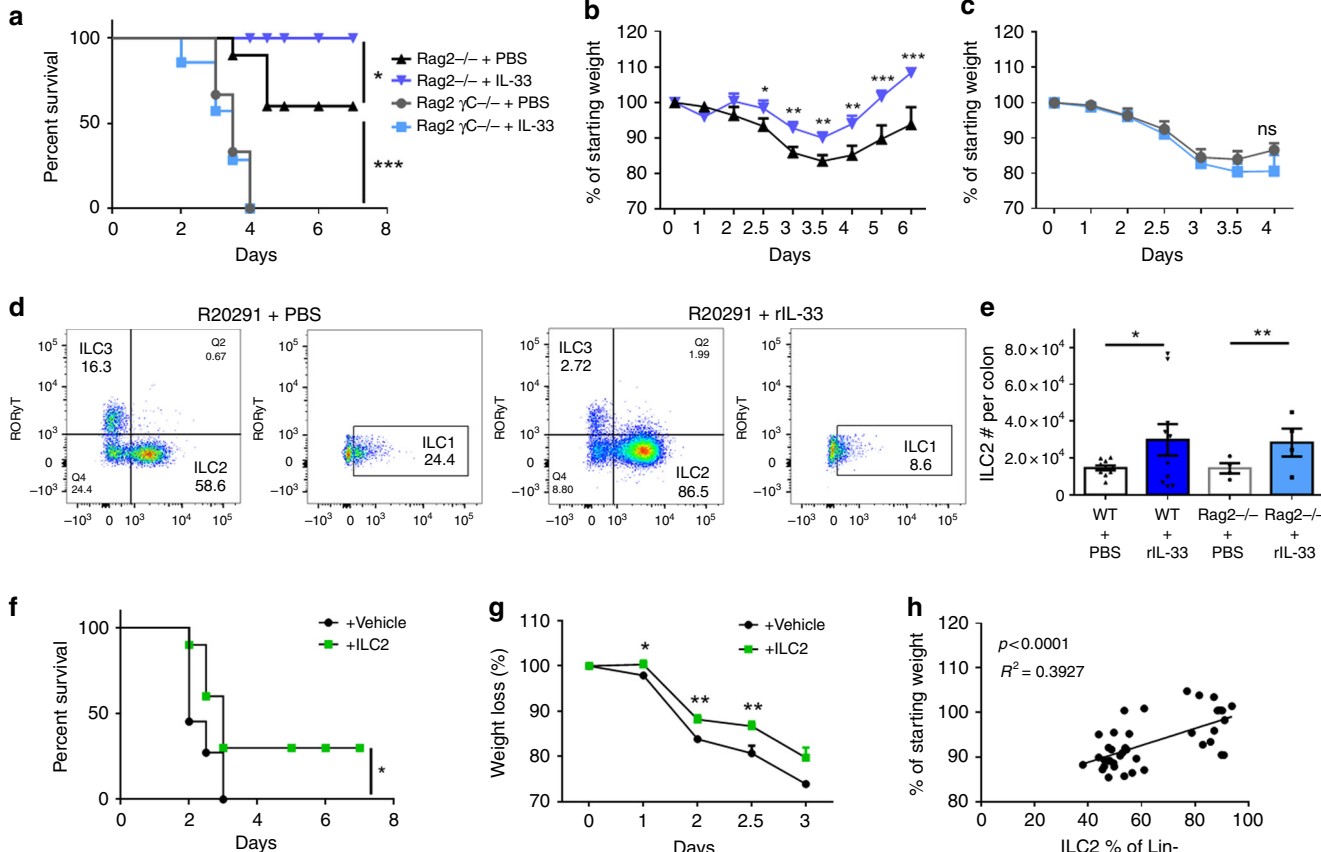

**Fig. 4** IL-33-ILC2 crosstalk is protective during *C. difficile* infection. **a–c** Rag2$^{-/-}$ and Rag2$^{-/-}$γc$^{-/-}$ were infected with R20291 and treated with or without IL-33. **a** Survival curves, (**b**) Rag2$^{-/-}$ weight loss, and (**c**) Rag2$^{-/-}$γc$^{-/-}$ weight loss after IL-33 treatment and infection. **d, e** Immune cell profiling of ILCs within the colon of IL-33-treated or untreated mice during R20291 infection. ILCs were subsetted into ILC2s (Lin- CD45$^+$ CD90$^+$ CD127$^+$ GATA$^+$ ST2$^+$); ILC1s (Lin- CD45$^+$ CD90$^+$ CD127$^+$Tbet$^+$), and ILC3s (Lin- CD45$^+$ CD90$^+$ CD127$^+$RORγT$^+$) by flow cytometry. **d** Representative dot plots of ILC subsets within the colon of PBS controls or IL-33-treated mice during infection (day 2). **e** Enumeration of the number of ILC2s in the colon of IL-33-treated C57BL6 or Rag2$^{-/-}$ mice during R20291 infection. **f, g** Adoptive transfer of ex vivo expanded and sorted purified ILC2s from uninfected IL-33 treated mice into naive Rag2$^{-/-}$γc$^{-/-}$ mice. **f** Survival curves and (**g**) weight loss after ILC2 transfer and R20291 infection. **h** Pearson correlation between the frequency of ILC2s and the percent weight loss during R20291 infection. **a, f** Comparison made by log rank test (**a**, $n = 6, 7, 10, 10$; **f**, $n = 11, 10$). (**b, c, e, g**) Comparison made by two-tailed Student's *t* test (**b**, $n = 8$; **c**, $n = 17,15$; **e**, $n = 10, 10, 4, 4$; **g**, $n = 10$). **h** Comparison made by Spearman correlation (**h**, $n = 39$). **a–c** The data representative of two independent experiments. **d–h** The data representative of three independent experiments. Statistical significance is demarked as *$P < 0.05$, **$P < 0.01$, and ***$P < 0.001$. Error bar indicates SEM

was independent of the adaptive immune system (Fig. 4e). We also noticed IL-33 treatment led to a significant reduction in the frequency of ILC1s, which canonically drive recovery during CDI[11], and a reduction in ILC3s (Fig. 4d; Supplementary Fig. 7c). The increase in ILC2s and corresponding decrease in ILC1s and ILC3s may be due to ILC plasticity within the intestinal lamina propria[34,35]. ILC2s isolated directly ex vivo from the colon during *C. difficile* infection were activated, as evidenced by their increased capacity to produce IL-13 compared with uninfected controls without further stimulation (Supplementary Fig. 8c). ILC2s isolated from IL-33-treated mice had 2.5-fold increase in IL-13+ ILC2s during infection compared with controls (Supplementary Fig. 8a, b). While ST2$^{-/-}$ mice had normal numbers of ILC2s in their colons, their ILC2s were dysfunctional as evidenced by their lack of IL-13 expression, indicating ST2$^{-/-}$ mice have reduced functional capacity during infection (Supplementary Fig. 8a, b, d, e). ST2$^{-/-}$ mice have deficiencies in type-1 CD4+ and CD8+ T-cell responses during viral infection[36,37], however, we did not see decreased frequencies or numbers of IFN-γ+ T cells during infection, likely due do the limited role for adaptive immunity during acute primary *C. difficile* infection

(Supplementary Fig. 8h, i)[11,38,39]. Furthermore, we did not see deficiencies in Tbet+ ILC1 numbers or alterations in their ability to produce IFN-γ in infected ST2$^{-/-}$ mice, together indicating that type-1 immunity is intact during ST2$^{-/-}$ infection (Supplementary Fig. 8f, g).

To determine whether ILC2s are sufficient to transfer protection during CDI, we adoptively transferred ILC2s from the mesenteric lymph nodes and colon of IL-33-treated mice and ex vivo expanded them in the presence of IL-33, IL-2 and IL-7 as described previously[40,41]. We then sort purified ST2+ ILC2s and transferred them into naive Rag2$^{-/-}$γc$^{-/-}$ mice prior to infection ($1 × 10^6$ ILC2s per mouse) (Supplementary Fig. 7a). We confirmed the presence of congenitally marked donor ST2+ ILC2s within the colon of recipient Rag2$^{-/-}$γc$^{-/-}$ mice (Supplementary Fig. 9b). Rag2$^{-/-}$γc$^{-/-}$ mice displayed 100% mortality during infection, however, ILC2 recipients had a 30% rescue in survival (Fig. 4f). Furthermore, ILC2 recipients were also protected from weight loss during their infection (Fig. 4g). Direct ex vivo transfer of ILC2s ($1 × 10^5$) from IL-33-treated mice into naive Rag2$^{-/-}$γc$^{-/-}$ mice also conferred protection from CDI-associated mortality (Supplementary Fig. 9a). Intestinal

ILC2s contribute to eosinophil accumulation and homeostasis within the gut[42,43]. Accordingly, ILC2 transfer enhanced colonic eosinophilia during *C. difficile* infection, demonstrating that transferred ILC2s were functional after trafficking into the colon (Supplementary Fig. 9c). Lastly, ILC2 frequency in the colon inversely correlated with weight loss during *C. difficile* colitis. (Fig. 4h). Collectively, these experiments reveal a previously unrecognized mechanism of protection from *C. difficile* colitis via IL-33 responsive ILC2s.

**IL-33 signaling is clinically relevant in human patients.** Next, we wondered whether the IL-33 signaling pathway was relevant in human *C. difficile* infection. Within the human intestine, IL-33 is expressed by pericryptal myofibroblasts and intestinal endothelial cells[44,45]. Systemic IL-33 in human serum was below the limit of detection by enzyme-linked absorbent assay (ELISA), however, subsequent immunohistochemistry staining revealed abundant anti-IL-33 staining of colon tissue biopsies from six human CDI+ patients (Fig. 5a, b). Blinded quantification of IL-33 staining in six CDI+ vs. six CDI− colon biopsies revealed a trend

toward increased IL-33 expression within the colon of patients with *C. difficile* (Fig. 5c). This complements our murine data that colonic IL-33 is upregulated in response to *C. difficile* (Fig. 1c).

We also detected the soluble IL-33 decoy receptor (sST2) at high levels in human serum by ELISA within a cohort of 167 CDI + patients. sST2 neutralizes the biological activity of IL-33 and in our murine CDI model, administration of the soluble IL-33 decoy receptor (sST2-FC) also increased weight loss and clinical symptoms, although no mortality occurred in either treatment or controls groups likely due to daily saline IPs during infection (Supplementary Fig. 10a–c). To determine whether sST2 expression is a biomarker of *C. difficile* disease severity in human patients, we stratified our CDI cohort into severe versus non-severe disease based on their white blood cell count. Patients with severe CDI were defined by a white blood cell count greater than or equal to 15,000 as others have defined previously[15,46]. We found that serum sST2 levels were higher in patients with severe CDI than non-severe CDI patients, and there was a significant correlation between sST2 expression and high-WBC count (Fig. 5d, e). A Kaplan-Meier curve and log-rank test

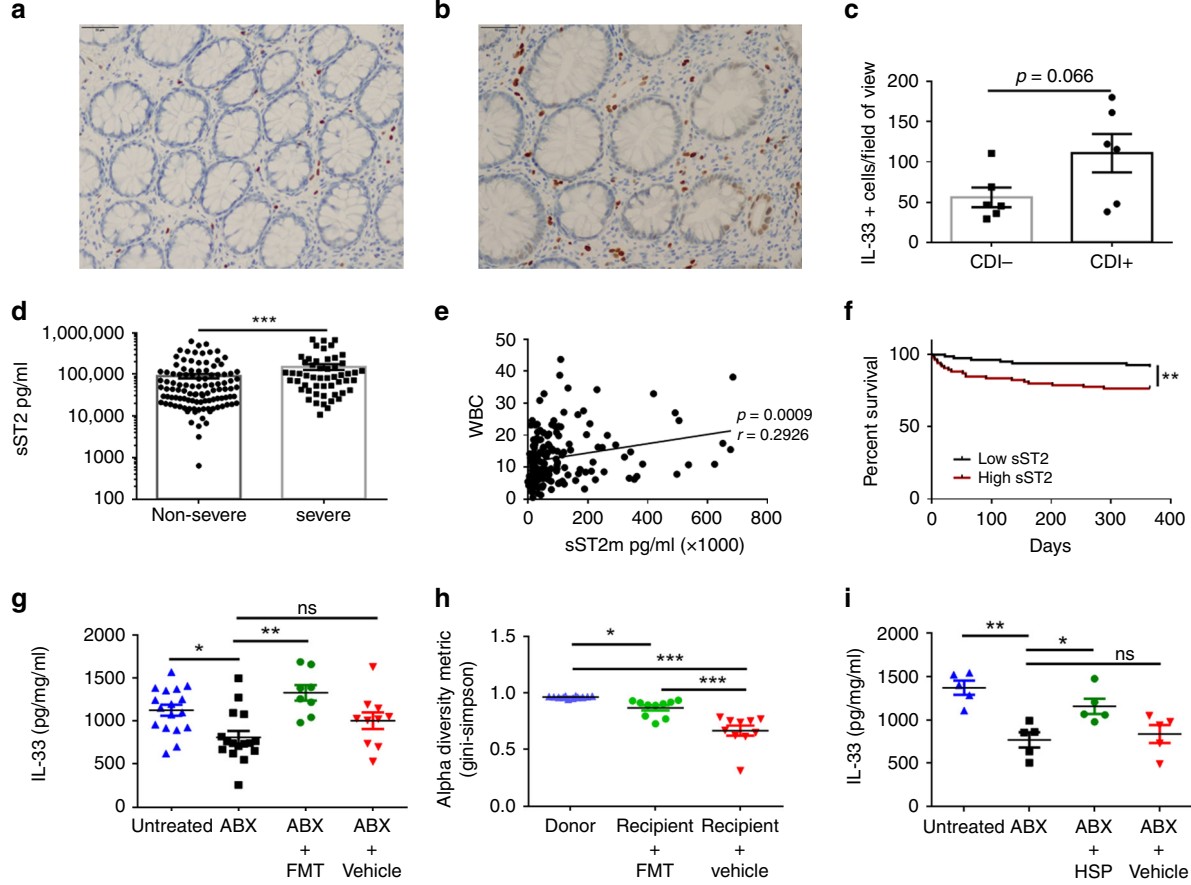

**Fig. 5** IL-33 signaling is dysregulated during human CDI and targetable with FMT or HSP therapy. **a–c** Immunohistochemistry staining of IL-33 from colon tissue biopsies of six CDI− patients and six CDI+ patients. **a** Representative image from a CDI− patient biopsy. **b** Representative image from a CDI+ patient biopsy. **c** Quantification of IL-33-positive cells per field view taken as an average of five blinded images from each patient biopsy and quantified on Image-J. **d–f** Analysis of systemic sST2 (IL-33 decoy receptor) in the serum of CDI patients by ELISA. **d** Patients were stratified based on their WBC into severe vs. non-severe CDI, and sST2 was assessed between the two groups. **e** Spearman correlation between WBC and sST2 concentration. **f** Survival curves of CDI+ patients stratified into high (sST2 > 55,000 pg/ml) or low sST2 (sST2 < 55,000 pg/ml). **g–i** Mice treated with a broad-spectrum antibiotic cocktail (ABX) were orally gavaged with a (**g**) murine FMT 2x or a (**i**) purified human spore preparation (HSP) isolated from a normal human donor and IL-33 protein within the cecal tissue was measured by ELISA. **h** Microbiota diversity in cecal contents of untreated donors and antibiotic-treated FMT recipients (Simpson index y axis). **c**, **d** Comparison made by Mann–Whitney test (**c**, n = 6, d, n = 109, 53). **e** Comparison made by Spearman correlation. **f** Comparison made by log rank test (**e**, **f**, n = 160). **g–i** Comparison made by ANOVA for multiple comparisons (**g**, n = 17,16, 8, 10; **h**, n = 12,10,10; **i**, n = 5). **g–h** Data representative of two independent experiments. Statistical significance is demarked as *P < 0.05, **P < 0.01, and ***P < 0.001. Scale bar is 50 μm. Error bar indicates SEM

revealed that patients with high sST2 expression (>median) had significantly increased mortality compared with patients with low sST2 expression (≤median) (Fig. 5f). Expression of these results as hazard ratios through a Cox regression model adjusting for age, gender, race, and Charson comorbidity score revealed that high sST2 expression was an independent prognosis factor for poor survival from *C. difficile* infection with a hazard ratio of 3.19 (Supplementary Table 1). Patient demographic distribution (age, gender, and race) is listed for patients with high ST2 and low ST2 expression (Supplementary Tables 2, 3). Together, this data indicate that the decoy receptor for IL-33, sST2, is a poor prognosis factor during human CDI.

**FMT rescues colonic IL-33 expression after antibiotics**. As FMT therapy is a highly effective treatment for severe relapsing CDI in humans, we wondered whether microbial therapy alters IL-33 abundance in the colon. We treated mice with a broad-spectrum cocktail of antibiotics to disrupt their protective microbial communities making them susceptible to CDI[8]. Twenty-four hours after antibiotic treatment, mice were orally gavaged with a mouse FMT isolated from ten age- and sex-matched wild-type donors. We found that mice treated with antibiotics had a significant decrease in their total colonic IL-33 protein expression (Fig. 5g). Oral gavage with a mouse FMT rescued IL-33 levels up to 10 days post treatment (Fig. 5g). Our murine FMT increased the diversity of antibiotic-treated recipients, making their communities more similar to donor non-antibiotic-treated controls with an expansion in the Firmicutes phylum (Fig. 5h; Supplementary Fig 11a, b). In addition, oral gavage with a human-derived fecal spore preparation (HSP) also significantly rescued IL-33 levels, indicating that a commensal or commensal-derived product is capable of regulating colonic IL-33 expression and is conserved between mouse and human microbial therapy (Fig. 5i). Taken together this data indicates that IL-33 signaling is relevant during human CDI and FMT therapy can rescue IL-33 expression in the colon after antibiotic-mediated depletion.

## Discussion
In this study, we investigated the host-derived mechanisms of immune protection during severe *C. difficile* via unbiased gut-derived transcriptomics. Here, we demonstrate that IL-33 signaling is a critical mechanism of intestinal protection during *C. difficile* colitis via the action of ILC2s. We see that IL-33 activates ILC2s in the large intestine and induces type-2 repair pathways that protect from toxin-mediated epithelial damage. Our findings have important relevance to intestinal health and infectious disease. The activity of IL-33 has centered around its contribution to protective "weep and sweep" defenses during helminth infection, however, this work expands IL-33's relevance to gut bacterial immunity. The key positioning of IL-33 at the epithelial interface enables it to integrate signals between the microbiota, pathogens, and the immune system. Our work highlights this sensing function, as colonic IL-33 was amplified in response to mouse and human *C. difficile* bacterial infection in addition to murine FMT or a human spore preparation. Thus, although IL-33 is constitutively expressed within the nucleus, its homeostatic expression is also regulated by microbial signals received from the gut lumen. Importantly, this work identifies IL-33 signaling to ILC2s as a therapeutically important pathway that is upregulated in the intestine during *C. difficile* colitis.

The primary virulence factors expressed by *C. difficile* are TcdA and TcdB, encoded from the pathogenicity locus (PaLoc)[47]. While both toxins can lead to disease, a recent study using isogenic mutants and three separate animal infection models, comprehensively demonstrated that TcdB is the major virulence

factor during in vivo infection and also the major driver of inflammation[48]. Hypervirulent Ribotype 027 isolates also express CDT toxin, which increases tissue pathology and mortality in murine infection models[5,6]. We demonstrate that endogenous and exogenous IL-33 protect from both epidemic R20291 [TcdA+ TcdB+ CDT+], and non-epidemic VPI 10643 [TcdA+ TcdB+ CDT−] isolates. This data indicate that IL-33-mediated protection is not restricted to CDT expressing strains, however is broader, also acting against TcdA and/or TcdB epithelial disruption. The relative contribution of each toxin toward IL-33 expression and release during *C. difficile* colitis requires further study. Given that IL-33 is released upon cell death, it is likely that toxin-mediated apoptosis or inflammasome-mediated pyroptosis are required for IL-33 release and activation of ILC2s during infection. IL-33 can be transcriptionally regulated during inflammation by sensing of pathogen-associated molecular patterns (PAMPS) via pattern recognition receptors (PRRs)[49]. For example, stimulation of TLR5 with microbial-derived flagellin induces expression of IL-33 mRNA in murine mucosal-associated dendritic cells and human corneal epithelial cells[50,51]. Another immune sensor, NOD1, which recognizes small peptides derived from peptidoglycan (PGN), is necessary for upregulation of IL-33 mRNA expression by gastric epithelial cells during *Helicobacter pylori* (*H. pylori*) infection[52]. Given that IL-33 is upregulated in response to both toxin expressing *C. difficile* and in response to a non-toxigenic human spore prep of Firmicutes, it is possible that toxin-independent factors contribute to IL-33 expression during *C. difficile* colitis. Specifically, *C. difficile* may increase IL-33 expression through PRR sensing of non-toxigenic signals, such as surface layer proteins, flagellin, and peptidoglycan, during infection[38].

Disturbances of the gut microbiota have long been associated with risk of developing *C. difficile* colitis. Gut microbial diversity provides colonization resistance against *C. difficile* through direct and indirect mechanisms of competition, production of metabolites, and induction of antimicrobial peptides[25,53,54]. Restoration of microbial diversity through FMT therapy is a successful treatment strategy in patients that fail to resolve their infection with traditional antibiotic therapy[4]. Previous studies have demonstrated a close relationship between IL-33 and the microbiota during murine IBD models with germ-free mice having reduced IL-33 levels in the small intestinal ilea during chronic ileitis[33,55]. Here, we show that murine FMT and a human-derived spore-based fecal preparation can rescue antibiotic-mediated depletion of colonic IL-33. The bacterial species or microbiota-derived signal regulating colonic IL-33 expression requires further investigation. The expression of PRRs, such as TLRs and NOD receptors and their subsequent sensing of microbial-derived patterns, is not restricted to pathogen encounter. In contrast, PRRs are expressed in the intestinal epithelium at steady state and are thus uniquely positioned to sense and interact with the luminal intestinal microbiota[56]. Thus, PRR sensing of commensal-derived antigens may also contribute to steady-state maintenance of IL-33 in the presence of the microbiota, similar to the MYD88-dependent antimicrobial peptide, RegIIIy, which aids in resistance to Gram-positive bacteria[57]. Thus, PRR signaling may not only contribute to IL-33 expression after *C. difficile* encounter but also after encounter with commensals within the gastrointestinal tract.

Bacterial-derived metabolites may also regulate colonic IL-33 expression in a similar fashion to epithelial-derived interleukin-25 (IL-25), a cytokine expressed by intestinal tuft cells and regulated by the organic acid succinate[10,58]. Like IL-33, IL-25 can protect the gut barrier during *C. difficile* colitis, indicating a shared role of barrier defense between these two epithelial-derived cytokines, possibly through ILC2s[10]. While IL-25 expression by tuft cells is

regulated by intestinal succinate, it is unknown whether microbial-derived metabolites can also regulate IL-33 expression. After murine FMT, we see an expansion in Firmicutes within the large intestine in addition to increased microbial diversity. Furthermore, our human spore cocktail was enriched in members of the Firmicutes phylum, including the families *Clostridiaceae*, *Erysipelotrichaceae*, *Eubacteriaceae*, *Lachnospiraceae*, and *Ruminococcaceae*. The majority of these families are members of the Clostridiales order which has been associated with CDI resistance through direct and indirect bacterial competition and metabolic alterations, including secondary bile acid and short-chain fatty acid production[59,60]. Thus, metabolite expression by Firmicute communities may also influence host immunity in the gastrointestinal tract via regulation of IL-33 expression.

ILCs have an important role in shaping gut immunity at homeostasis and during pathogen insult. While these cells lack antigen-specific receptors, they are tissue resident at the mucosal interface, enabling a quick and robust cytokine response. Recently, ILC1s were demonstrated to have a critical role in host defense during CDI while ILC3 involvement is limited[11]. ILC1s likely aid in localized antimicrobial defenses during CDI via IFN-γ production[11]. Our findings build off this work, demonstrating that IL-33 responsive ILC2s are also of high importance during CDI. ILC2s have tissue protective functions during murine IBD colitis and lung influenza, and function by protecting the epithelial barrier via the epidermal growth factor, amphiregulin[31]. Intestinal ILC2s also express the cytokine IL-13, which recently was demonstrated to act directly on the epithelium to drive goblet cell expansion and tuft cell hyperplasia aiding in parasite expulsion[26,27]. In line with this work, we saw that IL-33 activates ILC2s during *C. difficile* infection, increasing IL-13 production, increasing goblet mucin responses, and improving epithelial barrier function.

In addition, intestinal ILC2s constitutively express the eosinophil-stimulating cytokine IL-5 and have an essential role in eosinophil survival in the intestine[43,61]. In accordance with these findings, we see IL-33 treatment increases IL-5 levels during *C. difficile* infection, and adoptive transfer of ILC2s causes increased eosinophilia in the colon during *C. difficile* infection in addition to preventing CDI-associated weight loss and mortality. Our previous studies have demonstrated an important role for eosinophils during both infection with both hypervirulent and classical strains of *C. difficile*[6,10]. Specifically, IL-25 elicited eosinophils prevent CDI-associated mortality and eosinophil adoptive transfer is sufficient to promote survival[6,10]. Thus, ILC2s may act as upstream regulators of eosinophil function and barrier protection during CDI by integrating signals from the epithelium, such as IL-33 or IL-25, to promote repair. In line with this idea, recent studies have demonstrated a pivotal role for both IL-5 and eosinophils in human *C. difficile* infection with IL-5 being negatively correlated with disease severity and low peripheral eosinophil counts predicting poor CDI prognosis[16,17]. Thus, ILC2s may have a key regulatory role in countering *C. difficile* toxin-mediated epithelial damage and an overly robust antimicrobial response through their promotion of type-2 associated immunity.

The murine studies were validated in humans where we showed that high serum levels of the soluble IL-33 decoy receptor, sST2, were associated with more severe *C. difficile* colitis (defined by white blood cell count) and mortality. Previous findings have identified sST2 as a biomarker for mortality during human sepsis[62]. Murine models have demonstrated IL-33 can protect from sepsis and enhance antimicrobial neutrophil chemotaxis and also anti-inflammatory regulatory T cells and ILC2s[62–66]. A common theme may be that a robust inflammatory antimicrobial response is essential to control infection, yet this response must

be tightly regulated by immunosuppressive pathways to prevent tissue damage to the host. Furthermore, the infecting *C. difficile* strain dictates human clinical disease severity based on toxin status, and also causes changes to the host immune response. We previously demonstrated a detrimental role for CDT toxin expressed by NAP1/027 strains via its ability to kill eosinophils during infection[6]. Based on these previous findings and given IL-33's role in eosinophil survival, IL-33 may play a more important role in patients infected with a CDT expressing strain, and thus further assessment stratifying patient risk with CDT status, IL-33 expression, and eosinophil counts during human *C. difficile* infection will be important in future clinical studies.

IL-33 is considered an alarmin protein, transcriptionally regulated by PAMPS, and held within the nucleus until it is released upon cell death to activate rapid immunity. The acute nature of IL-33 signaling likely explains its importance very early on during *C. difficile* colitis in both our murine and human survival studies. The ability to regulate colonic IL-33 expression with certain spore-forming commensals adds to our knowledge regarding the immunomodulatory capabilities of FMT therapy. This work advances our fundamental understanding of the host immune response to *C. difficile*, demonstrating an important role for IL-33 responsive ILC2s in recovery and barrier protection during infection.

## Methods

**Mice and *Clostridium difficile* infection**. Work with animals complied with all relevant ethical regulations for animal testing and research and procedures were approved by the Institutional Animal Care and Use Committee at the University of Virginia (IACUC). Experiments were carried out using sex-matched 8–12-week-old C57BL6, ST2$^{-/-}$, Rag2$^{-/-}$, and Rag2$^{-/-}$γc$^{-/-}$ mice. C57BL6 were purchased from Jackson Laboratory, and ST2$^{-/-}$ mice[67] were obtained from Dr. Andrew McKenzie (Laboratory of Molecular Biology, Cambridge University, Cambridge, UK). Rag2$^{-/-}$ and Rag2$^{-/-}$γc$^{-/-}$ mice were purchased from Taconic Biosciences with an excluded flora. All animals were housed under specific pathogen-free conditions at the University of Virginia's animal facility. Bedding exchange every 2 days between ST2$^{-/-}$ and C57BL6 mice, or Rag2$^{-/-}$ and Rag2$^{-/-}$γc$^{-/-}$ mice, for a minimum of 3 weeks was conducted to equilibrate microbiota between strains. Mice were infected using a previously published murine model for CDI[6,10]. Three days prior to infection, mice were given an antibiotic cocktail in drinking water consisting of 45 mg/L Vancomycin (Mylan), 35 mg/L Colistin (Sigma), 35 mg/L Gentamicin (Sigma), and 215 mg/L Metronidazole (Hospira). Mice were then switched to regular drinking water and given a single IP injection (0.016 mg/g) of Clindamycin (Hospira) on day −1. On day 0, mice were orally gavaged with 1 × 10$^8$ CFU/ml of *C. difficile*. Mice were monitored twice daily over the course of infection and evaluated according to clinical scoring parameters. Scores were based on weight loss, coat condition, activity level, diarrhea, posture and eye condition for a cumulative clinical score between 1 and 20. Weight loss and activity levels were scored between 0 and 4 with four being greater than or equal to 25% loss in weight. Coat condition, diarrhea, posture, and eye condition were scored between 0 and 3. Diarrhea scores were 1 for soft or yellow stool, 2 for wet tail, and 3 for liquid or no stool. Mice were euthanized if severe illness developed based on a clinical score ≥ 14.

**Bacterial strains and culture**. Isogenic *C. difficile* strain R20291_*CdtB* was generated using the ClosTron system of insertional mutagenesis, and inactivation CDTb was confirmed by western blot by us and others[6,19]. To prepare the *C. difficile*, strains were plated onto BHI agar from frozen stocks and incubated at 37 °C overnight in an anaerobic work station (Shel Labs). A single colony was inoculated into the BHI medium and grown anaerobically overnight at 37 °C. The next day, cultures were spun for 1 min at 6000 × *g* and washed twice in anaerobic PBS, and the optical density was measured. For the strain VPI 10643 (ATCC 43244), 100 µl of overnight culture was subcultured for 5 h prior to optical density measurement. The culture density was adjusted in anaerobic PBS to 1 × 10$^8$ CFU/mL (R20291 strains) and 1 × 10$^5$ CFU/ml (VPI 10643 strain) and loaded into syringes. Each mouse received 100 µl (1 × 10$^7$ CFU for R20291 and 1 × 10$^4$ CFU for VPI 10643) of inoculum by oral gavage. *C. difficile* burden was quantified from cecal contents at day 2 of infection. Briefly, cecal contents were resuspended by weight in anaerobic PBS. Serial dilutions of cecal contents were plated on BHI agar supplemented with 1% sodium taurocholate, 1 mg/mL D-cycloserine, and 0.032 mg/mL cefoxitin (Sigma), and anaerobically incubated at 37 °C overnight followed by colony counts in triplicate. Toxins A/B and CDT were quantified using the ELISA *C. difficile* TOX A/B II kit from Techlab and CDT ELISA

(CDTb subunit detected) generously gifted from TechLab according to the manufacturer's instructions and normalized to stool weight.

**FITC-dextran gut permeability assay.** Mice were gavaged with 44 mg/100 g body weight of fluorescein isothiocyanate (FITC)–dextran solution (Sigma-Aldrich, # 46944-500MG-F). Four hours after gavage, mice were killed, and serum was collected. FITC dextran within the serum was detected on a spectrophotometer at 485/530 nm.

**Transcriptome microarray.** Mice were infected with R20291 (CDT+) or R20291_cdtb (CDT−) and whole-cecal tissue transcriptomic analysis was performed on day 3 post infection. R20291 and R20291_Cdtb RNA samples were processed by the Affymetrix Gene Chip® WT PLUS Reagent Kit and hybridized to the Affymetrix Mouse Gene 2.0 ST GeneChip®. The Affymetrix Mouse Transcriptome.CEL files were analyzed by the UVA Bioinformatics core. All preprocessing and analysis was done using R. Expression intensities were summarized, normalized, and transformed using Robust Multiarray Average algorithm[68]. Probesets not mapping to an Entrez gene were excluded. For examining differential gene expression, a linear model was fit with empirical-Bayes moderated standard errors using the limma package in R. The microarray analysis datasets are included in Table S1 comparing R20291 vs. R20291_Cdtb. Enriched pathways of the top upregulated transcripts (log FC > 0.5; p < 0.05) were created using the ConsensusPathDP database[21]. Enriched pathways of the top upregulated transcripts (log FC > 0.5) were also created using the Ingenuity Database[20].

**Tissue protein and transcript analysis.** IL-33, IL-4, IL-10, IL-1β, IL-6, and IL-23 were detected in cecal tissue lysates using the Mouse Duoset Sandwich ELISA kits (R&D) according to the manufacturer's instructions. The total cecal lysate was generated by removing the ceca and rinsing gently with 1x PBS. Tissue was bead beaten for 1 min and resuspended in 400 μl of lysis buffer I: 1 × HALT protease inhibitor (Pierce), 5 mM HEPES. Following mechanical tissue disruption, 400 μl of lysis buffer II was added: 1 × HALT protease inhibitor (Pierce), 5 mM HEPES, 2% Triton X-100. Tissue samples were incubated on ice for 30 min after gently mixing. Lysed samples were pelleted to remove tissue debris in a 5 min spin at 13,000 × g at 4 °C. Supernatant was collected, and the total protein concentration was measured by BCA assay according to the manufacturer's instructions (Pierce). Cytokine concentration was normalized to the total protein concentration. For IL-33 mRNA transcript analysis, cecal tissue was flushed with sterile PBS and immediately stored in RNA-later at −80 °C. Tissue was later processed using the RNeasy mini kit (Qiagen) + DNAse digestion. RNA was reverse transcribed with Tetro cDNA synthesis kit (Bioline), and IL-33 was amplified using the commercially available Taqman IL-33 primer/probe set (Applied Biosciences: Mm00505403_m1). Gene expression was normalized to HPRT and GAPDH housekeeping genes.

**Flow cytometry.** Colons were dissected longitudinally and rinsed in HBSS supplemented with 25 mM HEPES and 5% FBS. Epithelial cells were separated from the lamina propria via a 40-min incubation with gentle agitation in dissociation buffer (HBSS with 15 mM HEPES, 5 mM EDTA, 10% FBS, and 1 mM DTT) at 37 °C. Next, the lamina propria tissue was manually diced using scissors and further digested in RPMI 1640 containing 0.17 mg/mL Liberase TL (Roche) and 30 μg/mL DNase (Sigma). Samples were digested for 40 min at 37 °C with gentle shaking. Single-cell suspensions were generated by passaging samples through a 100 μM cell strainer followed by a 40 μM cell strainer (both Fisher Scientific). For intracellular cytokine staining, $1 \times 10^6$ cells/sample were incubated in complete media with Golgi block (RPMI Media 1640, 629 10% FBS, 2 mM L-glutamine, 100 U/ml penicillin, 100 μg/ml streptomycin, 5 mM 2-β-630 mercaptoethanol, BD GolgiPlug containing Brefeldin A) for 4 h prior to surface staining procedure. For staining, single-cell suspensions were obtained, and cells were stained with the following monoclonal antibodies: CD19 (115533, dilution 1/100), CD5 (100623, dilution 1/100), CD3 (100327, dilution 1/100), FcεRIα (134317, dilution 1/100), CD11c (117327, dilution 1/50), CD11c (117329, dilution 1/50), CD90 (105305, dilution 1/100), CD11b (101215, dilution 1/200), CD11b (101211, dilution 1/200), Ly6C (128005, dilution 1/100), CD45 (103115, dilution 1/200), Ly6G (127617, dilution 1/100), Tbet (505839, dilution 1/50), Ifny (505839, dilution 1/100) from Biolegend; ST2 (12-9333-82, dilution 1/100), RoryT (17-6981-82, dilution 1/50) from Ebiosciences; GATA3 (L50-823, dilution 1/50), SiglecF (552126, dilution 1/100), CD127 (566377, dilution 1/50) from BD; and IL-13 (61-7133-82, dilution 1/100) from Thermo Fisher. For surface staining, $1 \times 10^6$ cells/sample were Fc-blocked with TruStain fcX (BioLegend, #101320, 1/200) for ten minutes at room temperature followed by LIVE/DEAD Fixable Aqua (Life Technologies) for 30 min at 4 °C. Cells were washed twice in FACS buffer (PBS+ 2% FBS) and stained with fluorochrome conjugated antibodies for 30 min at 4 °C. Cells were washed and resuspended in Foxp3 Fix/Perm Working Solution (Ebiosciences, #00-5523-00) and incubated overnight at 4 °C. Cells were washed twice with permeabilization buffer and stained for for 45 min at room temperature. Flow cytometry was performed on an LSR Fortessa cytometer (BD Biosciences), and all data analysis performed via FlowJo (Tree Star Inc.).

**IL-33 Treatment and ILC2 adoptive transfer studies.** For IL-33 treatment, mice were intraperitoneally injected for 5 days prior to infection with 7.5 μg/ml (100 μl dose) of carrier-free, recombinant mouse IL-33 (Biolegend; Catalog #: 580504). For sST2 studies, mice were injected with 5 μg/mouse of recombinant ST2-FC fusion or FC control (R&D Systems, Catalog # 1004-MR-050). For ILC2 adoptive transfer studies, the colons, mesenteric lymph nodes, and spleens of CD90.1 + IL-33-treated mice were harvested into single-cell suspension as described above. Bulk ILCs were isolated by magnetic bead purification of lineage + cells (Miltenyi Lineage Cell Depletion Kit: 130-110-470). Cells were surface stained and sorted on the Influx Cell Sorter (BD Biosciences) based on Lin- ST2+ CD127+ CD45+ CD90.1+ expression. Sorted cells were immediately injected intraperitoneally into recipient CD90.2+ mice. For ILC2 expansion in vitro, ILCs purified from MLN and colon of IL-33-treated mice were cultured in complete medium (RPMI − 1640 + 10% FBS +2 mM glutamine, 100 U/ml penicillin, streptomycin 100 μg/ml) in the presence of IL-33 (50 ng/ml), IL-2 and IL-7 (10 ng/ml) for 4 days and sort purified by Lin- CD127+ CD45+ CD90+ ST2+ expression.

**FMT.** Fecal pellets from 10 age- and sex-matched C57BL6 mice were collected and immediately homogenized by vortexing in 1.5 ml of sterile anaerobic PBS[69]. In total, 100 μl of the supernatant was orally gavaged 24 h and 48 h post clindamycin injection. Human Spore Prep (HSP) is a research-grade product produced and provided by Seres Therapeutics and is comprised of Firmicutes spores fractionated from a stool specimen obtained from a healthy donor[70]. Spore content was determined by measuring spore colony-forming units per mL (sCFU/mL). In total, 200 μl of the human spore prep or vehicle control was administered 24 h post clindamycin IP into mice by oral gastric gavage.

**DNA extraction and 16S rRNA gene sequencing of murine FMT.** Generation of 16S rRNA gene amplicons from mouse cecal content was performed as previously described[69]. Briefly, DNA from cecal content was extracted using the Qiagen MagAttract PowerMicrobiome kit with the Eppedorf EpMotion liquid handling system. The V4 region of the 16S rRNA gene was amplified from each sample using a dual indexing sequencing strategy[71]. Samples were sequenced on the Illumina MiSeq platform using the MiSeq Reagent Kit V2 500 cycles (Illumina cat# MS102-2003) according to the manufacturer's instructions with modifications found in the Schloss Wet Lab SOP (https://github.com/SchlossLab/MiSeq_WetLab_SOP).

**16S sequence curation and analysis.** Raw sequence files have been deposited in the Sequence Read Archive database under the project PRJNA521980. All analyses were performed using R version 3.5.1. Sequences were analyzed using the R package DADA2 version 1.10.1, following the DADA2 pipeline v1.8 tutorial[72]. Taxonomy was assigned to amplicon sequence variants (ASVs) using the SLIVA rRNA database project, release 128[73]. The package phyloseq v1.26.1 was used to calculate both alpha and beta diversity, as well as generate the NMDS plots[74]. The package vegan was used to calculate the permutational multivariate analysis of variance (PERMANOVA)[75].

**Human spore prep sequencing and characterization.** Human Spore Prep was characterized by using 16S rRNA sequencing, sequenced to a depth of 22,500 operational taxonomic units (OTUs), at which point the rarefaction curves begin to plateau. The following families were detected by 16S sequencing: Clostridiaceae, Erysipelotrichaeceae, Eubacteriaceae, Lachnospiraceae, and Ruminococcaceae.

**Mouse and human histology and immunohistochemistry.** Mouse cecal tissue sections were fixed in Bouin's solution and transferred to 70% ethanol after 24 h. Sections were embedded in paraffin and stained with either haematoxylin and eosin (H&E) or PAS by the University of Virginia Research Histology Core. Scoring was conducted by two blinded observers. H&E tissue pathology scoring was conducted using a scale from 0 to3 for multiple parameters: epithelial disruption, submucosal edema, inflammatory infiltrate, mucosal thickening, and luminal exudates[76]. PAS+ goblet cells were counted and normalized to the number of crypts. For human anti-IL-33 staining, CDI positivity was assessed based on the presence of C. difficile toxins A/B in patient stool by ELISA. Human biopsies were obtained from the University of Virginia Biorepository and Tissue Research Facility. Tissues samples were provided from remnant surgeries, and researchers were blinded to patient identity. The University of Virginia Biorepository Core stained the human colon biopsy sections using a primary antibody directed against IL-33 (R&D, AF3625, 1/200). For human colon biopsies, five representative images were taken blindly of each tissue biopsy, and IL-33 positive cells were counted using the Image-J software particle analysis function.

**Detection of human sST2 in serum.** Work complied with all relevant ethical regulations for work with human participants and the study was approved by the University of Virginia Institutional Review Board for Health Sciences Research (IRB-HSR #16926). The Institutional Review Board for Health Sciences Research confirms that this project meets the criteria of research involving coded private information or biological specimens. According to the Office for Human Research Protections (OHRP) guidance, this project is considered to not involve human

subjects. Patients are diagnosed at UVA with *C. difficile* by a PCR test on stool: *C. difficile* nucleic acid amplification test (NAAT) GeneXpertSerum[77]. Samples were collected at the time of *C. difficile* diagnosis from 167 patients at the University of Virginia. Serum samples were stored at −80 °C until the time of use. Samples were thawed on ice and sST2 was quantified using R&D System Human ST2(IL-33R) Quantikine ELISA (Cat # DST200).

**Statistical analysis**. For murine work, survival curves were estimated using Kaplan–Meier method, and the survival difference between groups were tested for statistical significance using log-rank test. Comparisons between two groups were conducted using a two-tailed *t* test or a Mann–Whitney test when data were not normally distributed. For CDI human patients, comparisons between two groups were conducted using the two-sample t test or Mann–Whitney test as appropriate for continuous patient characteristics and using chi-square test for categorical variables. Seven observations with invalid time, censoring, or strata values were excluded from survival and Cox regression, and five observations with invalid WBC excluded from clinical severity Mann–Whitney test and spearmen correlation. Based on the distribution of sST2, patients were classified into low and high 50 percentiles of sST2 or tertiles of sST2. Survival differences between sST2 groups were assessed using log-rank test, and further evaluated in Cox regression adjusting for age, gender, race, Charlson comorbidity index. All statistical analyses were performed using GraphPad Prism software (GraphPad Software Inc., La Jolla, CA) and SAS 9.4 (SAS Institute, Cary, NC).

**Reporting summary**. Further information on research design is available in the Nature Research Reporting Summary linked to this article.

## Data availability

The microarray data generated in this study is available at the Gene Expression Omnibus under accession number GSE122013. Genes that met statistical significance in this study are listed in Supplementary Data 1. Raw sequence files for 16 s V4 sequencing have been deposited in the Sequence Read Archive database under the project PRJNA521980. Other data from the findings of this study are available from the corresponding author upon request. The source data underlying Figs. 1c, 2a–c, e–l, 3a, b, d–f, 4a–c, e–l, 5c–i, and Supplementary Fig. 1a–d; Figs. 2b–d, 3a–d, 4, 5b, c, 6b–l, 7c, 8, 9a, c, 10b, c are provided as a Source Data file.

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

## Acknowledgements

The authors thank TechLab, Inc. and Dr. David Lyerly and Mr. Matthew Lyerly for generously sharing CDT and Toxin A/B reagents. The authors thank Dr. Andrew McKenzie (Laboratory of Molecular Biology, Cambridge University, Cambridge, UK) for sharing ST2-deficient mice. We are grateful to Dr. Jonathan Kipnis and Dr. Tajie H. Harris for helpful assistance and expertise on IL-33. We also thank Drs. M. Rutkowski, A. Criss, J. Lukens, L. Erickson, and the Mann and Ramakrishnan laboratories for helpful discussions. We would also like to thank the University of Virginia Flow Cytometry, Histology, and Genome Analysis & Technology cores and the University of Michigan Microbial Systems Molecular Biology Laboratory. This work was supported by the US National Institutes of Health (5R01AI124214-02 and 1R21AI114734 to W.A.P., 1F31AI136421-01A1 to A.L.F., 5T32AI007496-24 to A.L.F., M.E.S. and E.L.B.), the Bill & Melinda Gates Foundation and by the Robert and Elizabeth Henske family. A.L.F. was also supported by the UVA Robert Wagner Fellowship.

## Author contributions

A.L.F. and W.A.P. designed all experiments. A.L.F. wrote the paper, performed all experiments and analyzed and interpreted data. M.M.S. helped with tissue extraction and processing, bacterial quantification, and invaluable advice. M.K.Y. aided in tissue collection, processing, mouse microbial therapy studies, IL-33 protein detection. E.L.B. aided with colon flow protocol development, tissue extraction and processing, and invaluable experimental guidance. M.E.S. aided in tissue collection and processing. P.P. conducted IL-33 histological staining on human colon biopsies. A.L.F., M.M.A. and J.Z.M. conducted human studies, including sample collection, quantification, and statistical analysis. A.L.F., S.D.T., and C.A.C. conducted microarray sample collection, processing and analysis. A.P.L. assisted with human spore cocktail preparation and experimental advice. J.L.L. analyzed and interpreted 16S rRNA sequencing data and wrote 16S rRNA sequencing methods section. M.M.S., M.K.Y., M.E.S., E.L.B., C.A.C. and J.L.L. assisted in the paper review and experimental advice. The University of Virginia

Genome Analysis & Technology Core conducted the Affymetrix GeneChip Microarray procedure and sample quality control. W.A.P. supported all aspects of this work.

## Additional information

**Competing interests:** W.P. is a consultant for TechLab, Inc. which manufactures diagnostic tests for CDI and A.F. and W.P. have filed a patent application on some of the work presented in the paper. The authors declare no other competing interests.

