## [Peer Review File · Nature Communications]

Editorial Note: This manuscript has been previously reviewed at another journal that is not operating a transparent peer review scheme. This document only contains reviewer comments and rebuttal letters for versions considered at Nature Communications .

IL-33 drives group 2 innate lymphoid cell-mediated protection during *Clostridium difficile* infection.

Frisbee and Petri et al.

Based on editor and reviewer comments the following changes have been made:

1. We have toned down language of microbiota-mediated protection in the manuscript. We have instead modified conclusions to state that IL-33 is upregulated by FMT and FMT has immunomodulatory roles in regulating IL-33 levels in the intestine. We have also changed the title of the paper to remove emphasis on microbiota-mediated protection.
2. We have revised the text to no longer focus primarily on CDT toxin in the abstract, introduction, and discussion.
3. We have provided more balanced discussion regarding IL-33 induction/release and the contribution of *C. difficile* virulence factors or non-virulence factors to this response within the discussion.

Editor Comments:

We therefore invite you to revise your paper one last time to address the remaining concerns of the reviewers. In particular, please make sure that the revised manuscript contains additional information on the human samples such as the toxigenic status of the clinical strains, provide a balanced discussion on IL-33 induction, IL-33 mediated protection, and role of different virulence factors in this regard (this includes e.g. the abstract that unnecessarily focuses on CDT), and tone down claims on protective role of microbiota in this regard (including the title). At the same time we ask that you edit your manuscript to comply with our format requirements and to maximise the accessibility and therefore the impact of your work.

Reviewer Comments:

Reviewer #1 (Remarks to the Author):

The authors have now addressed some of the concerns raised at first review phase. Specifically, toxin levels have been determined in infected mice, disease parameters have been included and additional microbiota sequencing has been performed, which has satisfied earlier concerns. I commend the authors for providing this new information.

I still have a few concerns about this paper, as detailed below:

Although de-emphasised, the *cdtb* mutant transcriptomics remain in the data set without inclusion of the complemented derivative, and there is no convincing rationale provided for conducting the *cdt* mutant transcriptomics compared to the WT. The other two toxins are far more important than CDT in damaging the host. Why were all three toxins not assessed in this context? If the point of the study was to find therapeutic immune targets relevant during severe

infection, toxin A and toxin B are far more relevant in this context (of which mutants are available in R20291). This is particularly important in the context of the new data suggesting that IL-33 mediated protection is not specific for CDT-expressing strains (although one other strain was used, specifically VPI10463 which produces toxin A and a different version of toxin B to R20291). Which virulence factor therefore induces this IL-33 response? It must be one of these toxins since non-toxigenic strains do not cause clinical disease. This has now become a very important mechanistic question to answer, which requires infection and transcriptomics with the toxin A and toxin B mutants of R20291, as well as the inclusion of these strains in downstream experiments.

To reiterate, the complemented version of the CDT mutant has still not been included in the analysis, which makes the transcriptomics based on this strain incomplete, as discussed in the first stage review of this paper.

Response: We thank the reviewer – we have included in the manuscript a more detailed discussion regarding both toxin A/B and their contribution to disease and potential release of IL-33 via apoptosis or pyroptosis. Furthermore, we have de-emphasized CDT in the abstract and introduction of the manuscript given that IL-33 mediated protection is not restricted to CDT+ strains. Furthermore, we have included additional discussion regarding the potential for non-toxigenic PAMPS to induce IL-33 mRNA expression via TLR signaling given that both non-pathogenic FMT and pathogenic *C. difficile* increase IL-33 levels.

The toxigenic status of the strains from infections derived from the 167 human serum samples is not stated. Since the toxins are the drivers of the immune response, and the toxigenic status of strains is highly variable, “CDI patients” is not a sufficient description. It is important to know the toxin status of these strains – were all of them toxin producers? Which diagnostic test was used to define these patients as CDI patients? No information is given.

Response: We thank the reviewer - we have included within the methods sections the method by which these patients were diagnosed for toxigenic *C. difficile*. (a PCR test on stool *C. difficile* nucleic acid amplification test (NAAT) GeneXpert)

Final items of note:

Line 517 – change VPI to VPI10463

Line 580 – change “classical strains” to “classical strain”, only one additional strain was used.

Response: Done, thank you.

Reviewer #2 (Remarks to the Author):

I thank the authors for the complementary work, which improved the manuscript. Additional experiments demonstrate that IL-33 mediated protection in the mouse model is not specific for CDT expressing strains, but is protective against 027 and non-027 (VPI) strains. These results strengthen the clinical studies in humans: increased colonic IL-33 expression in

CDI patients and especially high serum levels of the soluble IL-33 decoy receptor, sST2, associated with severe colitis. So, I suggest adding Figure S4 in the paper either alone or combined with Figure 2.

However, the results of the some experiments seem contradictory and raise some additional remarks and should be clarified:

The initial transcriptomic study demonstrated that IL-33 expression is upregulated in mice in response to infection with the CDT positive R20291 strain as compared to the CDT KO mutant. In contrast, IL-33 has a protective effect regardless of the infectious strain of *C. difficile*. It seems contradictory that IL-33 is overexpressed after infection with hypervirulent strains that are responsible for high morbidity and mortality in mice and that IL-33 reduced virulence (morbidity and mortality) of CDT+ or CDT- *C. difficile* strains. This duality should be discussed.

In addition, in the mouse model your results show that colonic IL-33 expression is lower in antibiotic-treated mice compared to non-treated mice and that FMT or gavage with human spores rescued IL-33 expression level. IL-33 level could be increased either by an intestinal pathogen such as *C. difficile* or by a normal microbiota. How could you explain these similar effects?

Response: We thank the reviewer for their helpful comments. We agree that toxin-independent mechanism may exist in IL-33 upregulation as IL-33 expression is increased by either pathogenic *C. difficile* or FMT. We have added additional discussion about how IL-33 may be upregulated in response to both pathogenic and nonpathogenic Clostridiales via Non-toxic factors and PAMPS conserved between the two. Furthermore, we have de-emphasized the importance of CDT and have a more balanced discussion of all three virulence factors.

Finally, You suggest that FMT efficacy to treat CDI could be linked to IL-33 signaling. However, this should be confirmed by additional experiments in the mouse model of CDI. This concern was not addressed. So, you should modulate conclusion and state that additional experiments in the mouse model of CDI will confirm the protective role of microbiota via IL-33 production.

Response: We thank the reviewer for their comments. We agree that we do not demonstrate that FMT is protective via IL-33, simply that IL-33 is upregulated in response to FMT therapy or a purified human spore prep. We have modified the conclusions and title of the paper to not conclude that FMT elicited IL-33 is protective from CDI. Instead, we state that IL-33 is potentially targetable or upregulated with FMT therapy.

Minor comments

Define abbreviations in the legend: Day instead of D; abx: antibiotic treated mice etc

Response: We have defined abbreviations in legends.

Please check Supplementary figure S6 and S7 that look similar?

Response: We have included the proper figure legend for Supplementary Figure 7

Reviewer #3 (Remarks to the Author):

Frisbee et al. revised manuscript addresses the majority of the critiques brought up by this reviewer. However, a few comments remain:

1. The data in Fig S8B compares ILC2 IL13 expression between WT, rIL-33 & ST2 KO mice. While figure S8A does have the uninfected controls, the data as currently presented does not support the statement on line 230-232 "GATA3+ ILC2s isolated directly ex vivo from the colon during *C. difficile* infection were activated, as evidenced by their increased capacity to produce IL-13 compared to uninfected controls without further stimulation (Fig S8A-B)."

Response: We thank the reviewer for this comment. We have included an additional comparison in Supplementary Figure 8C between uninfected mice and infected mice. We see a significant upregulation of IL-13+ ILC2s during *C. difficile* infection relative to uninfected controls.

2. The authors included the ILC gating strategy in Figure S7. However, the Figure Legend is for Fig S6. Please place in the proper figure legend that describes the data in Fig. S7.

Response: We have included the proper figure legend for Figure S7. Thank you.